# The Diagnosis, Pathophysiology, and Treatment of Chronic Hepatitis E Virus Infection—A Condition Affecting Immunocompromised Patients

**DOI:** 10.3390/microorganisms11051303

**Published:** 2023-05-16

**Authors:** Satoshi Takakusagi, Satoru Kakizaki, Hitoshi Takagi

**Affiliations:** 1Department of Gastroenterology and Hepatology, Kusunoki Hospital, 607-22 Fujioka, Fujioka 375-0024, Gunma, Japan; satoshi.takakusagi@gmail.com; 2Department of Clinical Research, National Hospital Organization Takasaki General Medical Center, 36 Takamatsu-cho, Takasaki 370-0829, Gunma, Japan

**Keywords:** chronic hepatitis E, immunocompromised, immunocompetent, HEV RNA, pathogenesis, ribavirin

## Abstract

Hepatitis E is a zoonosis caused by hepatitis E virus (HEV), which was first discovered 40 years ago. Twenty million HEV infections worldwide are estimated each year. Most hepatitis E cases are self-limiting acute hepatitis, but the virus has been recognized to cause chronic hepatitis. Following the first case report of chronic hepatitis E (CHE) in a transplant recipient, CHE has recently been identified as associated with chronic liver damage induced by HEV genotypes 3, 4, and 7—usually in immunocompromised patients such as transplant recipients. In addition, patients infected with HIV and those receiving chemotherapy for malignancy, along with patients with rheumatic disease and COVID-19, have recently been reported as having CHE. CHE can be easily misdiagnosed by usual diagnostic methods of antibody response, such as anti-HEV IgM or IgA, because of the low antibody response in the immunosuppressive condition. HEV RNA should be evaluated in these patients, and appropriate treatments—such as ribavirin—should be given to prevent progression to liver cirrhosis or liver failure. While still rare, cases of CHE in immunocompetent patients have been reported, and care must be taken not to overlook these instances. Herein, we conduct an overview of hepatitis E, including recent research developments and management of CHE, in order to improve our understanding of such cases. The early diagnosis and treatment of CHE should be performed to decrease instances of hepatitis-virus-related deaths around the world.

## 1. Introduction

Hepatitis E is an inflammation of the liver caused by infection with the hepatitis E virus (HEV), and it is a major cause of acute viral hepatitis worldwide [1]. The virus is usually transmitted via the fecal–oral route, principally via contaminated water [1]. Ingestion of undercooked animal meat (including animal liver, and particularly pork) also causes the infection. The World Health Organization (WHO) estimates that there are 20 million HEV infections worldwide each year, leading to 3.3 million symptomatic cases and 44,000 deaths [2,3]. It accounts for 3.3% of mortalities due to viral hepatitis [2]. Hepatitis E is found worldwide, but the disease is most common in East and South Asia [2].

The clinical features of hepatitis E are generally mild and include elevated liver enzymes, jaundice, and nonspecific symptoms such as appetite loss, malaise, and abdominal pain. These symptoms are often indistinguishable from those experienced during other liver illnesses and typically last 1–6 weeks. Although almost all cases are self-limiting, cases of chronic hepatitis E (CHE) infection have been reported in immunosuppressed people—particularly organ transplant recipients on immunosuppressive drugs [4]. In rare cases, acute hepatitis E can be severe and result in fulminant hepatitis or acute liver failure. Although such chronic infections of HEV or fulminant hepatitis are rare, this may result in a life-threatening illness. Chronic infection with HEV can lead to liver cirrhosis or hepatocellular carcinoma [5,6].

We therefore summarize and discuss current epidemiological and clinical findings concerning chronic HEV infection. We also discuss the diagnosis and treatment of chronic HEV infection.

## 2. Hepatitis E Virus Overview

HEV is classified into the family *Hepeviridae* [7]. HEV is a positive-sense, single-stranded, non-enveloped, icosahedral RNA virus of 32 to 34 nm in diameter [7]—one of five known human hepatitis viruses (A–E). The encapsulated 7.2 kb genome of HEV resembles eukaryotic mRNA with a 5′7-methylguanylate cap and a 3′ poly(A)tail and consists of three major open reading frames (ORFs; ORF1-3) [7,8,9]. ORF1 encodes nonstructural proteins, ORF2 encodes the viral capsid proteins, and ORF3 encodes a protein with an unknown function [7].

The classification is based on the nucleotide sequences of the genome [7], and HEV has eight different types (genotypes 1–8). Among these eight genotypes, genotypes 1, 2, 3, 4, and 7 infect humans [10]. Genotypes 1 and 2 have been found only in humans and are often associated with large outbreaks and epidemics in developing countries with poor sanitation conditions [11]. Genotypes 3, 4, and 7 circulate in several animal species, including pigs (HEV-3 and -4), rabbits (HEV-3), wild boars (HEV-3, -4, -5, and -6), mongooses (HEV-3), deer (HEV-3), yaks (HEV-4), and camels (HEV-7 and -8), without causing any disease, and have been responsible for sporadic cases of hepatitis E in both developing and industrialized countries [12,13]. It is spread mainly by the fecal–oral route due to contamination of water supplies or food [14]. Figure 1 illustrates the different routes of transmission of HEV to humans. Although direct person-to-person transmission is uncommon [14], mother-to-child transmission, blood transfusion, and organ transplant are reported routes of HEV transmission [15,16]. In contrast to genotypes 1 and 2, genotypes 3, 4, and 7 cause sporadic cases that are thought to be contracted zoonotically—via direct contact with animals, or indirectly from contaminated water or undercooked meat [14].

The virus is shed in the stool of infected individuals and enters the human body through the intestine. The incubation period following exposure to HEV ranges from 2 to 10 weeks, with an average of 5 to 6 weeks. The infected individuals excrete the virus beginning from a few days before to 3–4 weeks after the onset of the disease. In areas with high disease endemicity, symptomatic infection is most common in young and middle-aged adults aged 15–40 years. In these areas, although infection does occur in children, it often goes undiagnosed, as they typically have no symptoms or only a mild illness without jaundice.

In general, acute HEV infection is relatively asymptomatic or mildly symptomatic. Acute icteric hepatitis is seen in around 5–30% of patients infected by HEV [10]. Typical signs and symptoms of hepatitis include an initial phase of a mild fever, reduced appetite, nausea and vomiting lasting for a few days, abdominal pain, itching, skin rash, joint pain, and jaundice. These symptoms are often indistinguishable from those experienced during other liver illnesses and typically last 1–6 weeks. Viremia precedes jaundice, and the virus is also shed in the feces. Diagnosis includes serological antibody measurement and HEV RNA detection. As serological diagnostic markers, the IgA, IgM, and IgG antibodies are detected after HEV infection. IgA and IgM antibodies are elevated during the acute phase of HEV infection, and then they decrease or become negative during the convalescent and healing phases. Therefore, in the diagnosis of hepatitis E, IgM or IgA HEV antibodies are detected in this acute-phase serum. IgA and IgM antibodies become negative 3 to 6 months after onset, and IgG antibodies are considered to be a marker for a history of HEV infection because they remain positive for a long period of time. The IgA-class HEV antibody measurement system has higher specificity than the IgM-class HEV antibody measurement system and is used for diagnosis mainly in Japan [17]. Diagnosis is also possible by detecting HEV RNA in the patient’s serum or feces. HEV RNA testing is highly sensitive and specific. Since the amount of HEV in the blood usually reaches a peak before the onset of symptoms, it rapidly declines thereafter and becomes negative. Therefore, it is important to secure specimens as soon as possible after the onset in order to detect HEV RNA. It can be detected in feces for approximately 10 to 29 days after onset and in serum for approximately 7 to 40 days after onset. 

Generally, HEV genotypes 1 and 2 cause more severe acute hepatitis in comparison to HEV genotypes 3 and 4 [18]. Nonetheless, HEV genotypes 3 and 4 may lead to severe acute HEV infections in older men and acute-on-chronic liver failure (ACLF) in patients with chronic liver disease. Although most cases are self-limiting, cases of CHE infection have been reported in immunosuppressed people—particularly organ transplant recipients on immunosuppressive drugs.

In rare cases, acute hepatitis E can be severe and result in fulminant hepatitis or acute liver failure. Pregnant women with hepatitis E—particularly those in the second or third trimester—are at an increased risk of acute liver failure, fetal loss, and mortality. Up to 20–25% of pregnant women can die if they contract hepatitis E in the third trimester [15]. In addition to signs of acute infections, adverse effects on the mother and fetus may include preterm delivery, abortion, stillbirth, and neonatal death [19]. The pathological and biological mechanisms behind the adverse outcomes of pregnancy infections remain largely unclear. Increased viral replication and the influence of hormonal changes on the immune system are currently thought to contribute to worsening of the course of infection [20]. Furthermore, studies showing evidence for viral replication in the placenta or reporting the full viral life cycle in placenta-derived cells in vitro suggest that the human placenta may be a site of viral replication outside the liver [20]. The primary reason for the severity of HEV in pregnancy remains enigmatic.

HEV has been proposed as an overlooked infectious trigger of ACLF. Manka et al. [21] performed a retrospective analysis of 80 acute liver failure cases in a single center from Germany. Among all patients, eight were HEV-RNA-positive and had supporting clinical findings of acute HEV infection; however, half of them initially received an erroneous diagnosis of drug-induced liver injury. Another study from the United States showed that a minority of suspected cases of drug-induced liver injury were actually caused by HEV [22]. In cases of superinfection of HEV on chronic liver disease, HEV accelerated disease progression and increased the rates of liver failure and mortality in patients with chronic hepatitis B [23] and other forms of chronic liver disease [24]. 

Infection with HEV can also lead to problems in other organs. Although the precise relationship is not entirely clear, acute pancreatitis, neurological complications (e.g., Guillain–Barré syndrome, neuralgic amyotrophy, acute transverse myelitis, and acute meningoencephalitis), glomerulonephritis with nephrotic syndrome, and thrombocytopenia have been reported in patients with HEV infection [25,26,27]. Acute pancreatitis is associated not only with HEV but also with other hepatitis viruses (A, B, and C). HEV-associated acute pancreatitis is usually resolved with supportive care and is mild–moderate in severity [26]. Guillain–Barré syndrome is one of the most frequently published extrahepatic complications of HEV infection, and it can occur after both acute and chronic infections of various HEV genotypes [28]. In a study from the United Kingdom and France, more than 5% of patients with HEV-3 infection had neurological complications during follow-up [27]. HEV can cause glomerulonephritis in both immunocompetent and immunosuppressed patients [29,30]. In HEV-induced glomerulonephritis, immune-mediated mechanisms presumably play an important role similar to hepatitis C virus (HCV)-associated glomerulonephritis, in which immune complexes accumulate in the glomerular tissue. HEV clearance was achieved either by therapy or spontaneous improvement of the renal functions and proteinuria levels [31]. Although severe thrombocytopenia was observed in case reports, thrombocytopenia associated with HEV infections is generally not severe and does not require any specific treatment [32]. The extrahepatic manifestations of HEV infections can be treated by either ribavirin or immunosuppressive medications, such as corticosteroids. Before deciding treatment for these manifestations, the main mechanism of extrahepatic manifestation in question should first be determined. These extrahepatic manifestations are supposed to be mainly mediated by immunological mechanisms or the direct viral (cytopathic) effect of HEV. Therefore, treatment should be chosen according to the main pathophysiological mechanisms of extrahepatic manifestations.

Hepatitis E due to genotypes other than 1 and 2 is thought to be a zoonosis, in that animals are thought to be the primary reservoir; indeed, deer and swine have frequently been implicated [33]. Domesticated animals have been reported as a reservoir for HEV as well, with some surveys showing infection rates exceeding 95% among domestic pigs [34]. Replicative virus has been found in the small intestine, lymph nodes, colon, and liver of experimentally infected pigs. Transmission after consumption of wild boar meat and uncooked deer meat has been reported as well [33]. However, the rate of transmission to humans by this route is still unclear [35].

Prevention is the most effective approach against infection. At the population level, transmission of HEV can be reduced by maintaining quality standards for public water supplies and establishing proper disposal systems for human feces. On an individual level, the risk of infection can be reduced by maintaining hygienic practices and avoiding consumption of water and ice of unknown purity. A vaccine to prevent HEV infection has been developed and is licensed in China [2] but is not yet available elsewhere.

## 3. Chronic Hepatitis E (CHE)

### 3.1. Definition of CHE and Its Historical Occurrence

Although acute hepatitis E has usually been recognized as having a favorable prognosis without prolonged viremia, as explained in the Introduction section of this article, HEV genotypes 3, 4, and 7 can cause chronic infection in immunocompromised individuals [10,36,37]. Most cases of CHE have been caused by genotypes 3 and 4, with only one case report of genotype-7-induced CHE [35], which usually infects camels and is not expected to infect humans. According to this case report [35], a liver transplant recipient from the Middle East who regularly consumed camel meat and milk was chronically infected with genotype 7 HEV [37]. Chronic HEV infection is usually defined as the persistence of HEV replication for six months [38]. In contrast, CHE is defined as the presence of persistently elevated liver enzyme levels and a viremic status for more than three months [39].

However, in a small number of cases, spontaneous clearance has been observed between three and six months [40]. A quarter of a century has passed since the discovery of HEV [36] (Figure 2). As immunocompromised causes of persistent HEV infection, organ transplantation, HIV infection, and chemotherapy have been reported (Figure 2), as explained further in the latter part of this section.

### 3.2. Pathogenesis of CHE

Chronic HEV infection can occur in immunocompromised hosts in whom the adaptive immune system is impaired. Regarding the clinical course of transplant patients, approximately 70% of solid organ transplant patients infected with HEV develop chronic hepatitis [41]. Individual differences in the immune response to HEV exist, as not all transplant patients treated with the same immunosuppressant develop chronic HEV infection; indeed, many anti-HEV-antibody-positive and HEV-RNA-positive organ transplant patients receive the same immunosuppressant regimen. HEV-specific CD4 and CD8 T-cell responses were reported to be undetectable in chronically HEV-infected patients but manifested after viral clearance [42]. In a unique macaque model, HEV clearance was accompanied by a neutralizing antibody response and liver infiltration of functional HEV-specific CD4 and CD8 T cells [43]. Innate immune responses to hepatitis E viral proteins translated from several ORFs have been demonstrated, and the persistence of HEV may be caused by these individual immunological differences [44].

As an immunosuppressant used in transplant patients, the use of tacrolimus rather than cyclosporine and a low lymphocyte count at diagnosis are associated with chronic infection after exposure to HEV [45]. A subsequent case report documented persistent HEV infection in a patient with HIV [46], but another study revealed that HIV-positive patients were not a risk group for HEV coinfection [47]. This may explain why chronic HEV infection is rare in patients with HIV, who typically experience a progressive loss of CD4 T cells in isolation. Patients with cancer receiving chemotherapy and/or immunotherapy [48], as well as those with autoimmune diseases (e.g., rheumatic diseases) receiving immunosuppressive or immunomodulatory therapies [49], are also at risk of developing CHE.

It should be noted that CHE in immunocompetent patients has rarely but occasionally been reported in several countries, including the USA [50], Germany [51], France [52], China [53], and Japan [54]. These cases were not organ-transplant- or cancer-related; instead, most had been administered immunosuppressants such as glucocorticoids and mycophenolate mofetil, except for the Chinese case [53]. This patient had a purely immunocompetent condition, rarely complicated with chronic HEV infection [53]. The Japanese case was also administered prednisolone and azathioprine, but the presence of a long-lasting high titer of HEV RNA before the administration of these drugs resulted in her diagnosis with CHE despite her immunocompetent condition and older age (late 70s) with underweight status (body mass index: 19.8 kg/m^2^) and without any immunocompromised comorbidities [54]. These two patients seemed to be immunocompetent but, nevertheless, developed CHE. As shown in the Japanese case [54], the patient’s age (>70 years) and low body mass index (<20 kg/m^2^) correspond to a severely undernourished status according to the Global Leadership Initiative on Malnutrition (GLIM) criteria [55]. Immune responses are reportedly decreased in undernourished elderly individuals, along with the CD4+ population [56]. Protein–energy malnutrition (PEM) and aging lead to severe immunodeficiency in the elderly population, affecting not only specific immunity (B and T lymphocytes) but also nonspecific immunity (polymorphonuclear cells and monocytes). PEM patients release fewer monokines, which leads to reduced stimulation of lymphocytes, the functions of which are already diminished. PEM patients are therefore unable to raise effective immune reactions [56].

It has been reported that HEV clearance is mainly dependent on T-cell responses, because lymphocyte subset counts—mainly CD4+ lymphocyte counts—have been found to be significantly lower at the diagnosis of HEV infection in patients who develop chronicity in comparison to those who clear the virus within 6 months [45]. In patients with HCV infection, HCV-specific CD4+ and CD8+ T lymphocytes play a crucial role in the eradication of the HCV in the liver [57]. Another study demonstrated that the number of IFN-γ–producing HCV-specific CD8+ T cells during the first 6 months after the onset of disease was associated with the eradication of HCV infection [58]. HEV eradication was also dependent on the IFN-γ sequence of HEV-specific CD8+ T cells [59]. Although the frequency of persistent viremia and the route of infection are quite different between HCV and HEV, both are hepatotrophic RNA viruses, and the immunological approach to HCV could be analogously applied for HEV. COVID-19 has also been reported to cause CHE [60], but no additional convincing reports regarding the relationships between COVID-19 and chronic HEV infection have been documented recently. Further studies are needed to elucidate the mechanism of CHE and to document the many hidden cases of undiscovered CHE.

### 3.3. The Diagnosis and Clinical Course of CHE

Anti-HEV antibodies are often undetectable in immunosuppressed patients who are chronically infected with the virus (Table 1) [54,61,62,63,64]. Variable immunoglobulin isotype responses to HEV are described in Table 1. To diagnose HEV infection based on an antibody response, HEV IgM has been used worldwide, while HEV IgA is mainly used in Japan [17]. However, these antibody responses have demonstrated limited sensitivity, especially in immunocompromised situations [1,2], as shown in Table 1. The nucleic acid amplification technique-based detection of viral RNA in blood and/or stool samples is the only reliable method for making a diagnosis. Chronic HEV infection is defined by the detection of viral RNA for more than three months. The viral load calculated from the quantification of HEV RNA is also used to evaluate the treatment response [38]. The diagnosis is usually only considered when infection is clinically suspected, but this is clearly insufficient to fully recognize the burden of HEV infection in immunocompromised patients. Figure 3 shows the diagnostic process in cases of hepatitis E infection. HEV RNA is not always evaluable everywhere in the world, but recent advancements in polymerase chain reaction (PCR) for the diagnosis of COVID-19 have improved the commercial availability. Clinicians must therefore not overlook the possibility of chronic HEV infection based on one-time-negative anti-HEV antibody findings.

Questions have been raised as to whether or not structured systematic screening of HEV infection for transplant recipients living in endemic areas should be considered [65], but further research—especially cost-effectiveness analyses—will be required to address this issue [36]. 

CHE may progress to liver cirrhosis [36,66], but the hepatocarcinogenic potential of persistent HEV infection has yet to be clarified, due to the rarity of HCC in cases of advanced CHE [5,6]. Epidemiological studies have shown both a positive and a negative relationship between HEV and HCC [6]. A Chinese group showed that HEV infection was not an independent risk factor for HCC, but that HBV and HEV coinfection might be positively associated with the development of HCC [67]. Another group in Cameroon demonstrated that HCC patients had a higher seroprevalence of HEV IgG in comparison to patients with chronic liver disease [68]. Superinfection of hepatitis B or C and HEV may enhance hepatocarcinogenesis; however, at this time, we do not consider simple HEV infection to be sufficient to induce HCC. Large amounts of experimental data have also been accumulated on the relationship between HEV and hepatocarcinogenic pathways such as angiogenesis, apoptosis, oxidative stress, and chronic hepatic inflammation with fibrosis [69], which are seen in CHE, and which could cause HCC [5,6]—the same as it is caused by chronic HCV and HBV infection, habitual alcohol intake, genetic disorders such as hemochromatosis, and other conditions [69,70]. At any rate, further evidence must be accumulated for HEV to be accepted as a cause of HCC.

### 3.4. Prevention and Treatment for CHE

The hope for better prevention of HEV infection lies in the availability of safe and effective vaccines [36]. The only approved vaccine (Hecolin) was licensed in China in 2011. It is well tolerated and effective in the prevention of HEV genotypes 1 and 4 in the general population, with long-lasting protective immunity [71,72]. Although these findings cannot be directly applied to HEV genotype 3, immunization with Hecolin in a rabbit model has been shown to confer full protection against HEV genotype 3 [73]. In addition, the World Health Organization (WHO) calls for studies to evaluate the safety and immunogenicity of Hecolin in special populations such as immunocompromised patients and pregnant women, and an ongoing phase IV trial is testing protection in pregnant women in Bangladesh [74]. On the other hand, the vaccination with Hecolin in rabbits prior to the administration of immunosuppressive medication fully protected them against HEV genotypes 3 and 4, whereas only partial protection was achieved when the animals were already receiving immunosuppressants [75]. From a clinical practice perspective, one could argue in favor of preferentially vaccinating patients on the waiting list for organ transplantation [36].

Regarding the treatment for CHE (Table 2), in immunocompromised cases, the first-line therapeutic option for chronic HEV infection is a reduction in immunosuppressive therapy [76]. HEV clearance was reportedly observed in 30% of solid organ transplant patients with chronic HEV infection after reducing immunosuppressive therapies that principally targeted T cells [41]. It has been recommended that if HEV infection is not cleared within three months after starting dose reduction of immunosuppressants, antiviral therapy should be considered [41].

Ribavirin (RBV) has been reported to show excellent efficacy and safety in treating CHE. In a multicenter retrospective study including 59 solid organ transplant recipients treated with RBV, a sustained virologic response (SVR)—defined as an undetectable serum HEV RNA level for at least 6 months after the cessation of medication—was achieved in 78% of initial treatment cases, 85% of overall cases (including retreatment cases [77]), and 83% of liver transplant recipients [78]. The exact mechanism of RBV on HEV clearance remains unclear. RBV could act both directly on viral replication and as an immunomodulatory agent [77]. The effect of RBV on HEV was demonstrated in another mechanism through natural killer (NK) cells [79]. An in vitro study demonstrated that RBV has an immunomodulatory effect on the IL-12R pathway of NK cells via tyrosine kinase (TYK)-2. This subsequently leads to an enhanced IFN-γ response as an additive antiviral effect in the context of HEV infection [79].

Although the optimal dose of RBV for CHE has not yet been established in both immunocompromised and immunocompetent cases, RBV was initiated at a median dose of 600 mg/day and achieved high rates of SVR, and the dose was often adjusted in response to adverse events such as RBV-induced anemia [77,80]. It was also reported that no statistically significant difference in the SVR rate was observed between patients treated for three months or less and those who received RBV for more than three months [77]. In contrast, however, 6 of 10 relapsed patients who had been initially treated for 3 months were retreated with RBV for 6 months, and an SVR was achieved in 4 of these patients [77]. Therefore, three months of RBV administration is considered appropriate as the initial treatment, but it should be extended to six months at retreatment. Regarding the mechanism underlying the effect of RBV on HEV, RBV inhibits HEV replication through the depletion of guanosine triphosphate (GTP) [81]. It was also recently shown that although RBV increases viral heterogeneity, RBV-induced metagenesis seems to be reversible after therapy is stopped [82]. Another study showed that an RBV-failure-associated mutation (Y1320H) in the RNA-dependent RNA polymerase of HEV genotype 3 enhanced viral replication in a rabbit HEV infection model [83]. A more precise investigation is needed to clarify the effect and the reason for failure of the RBV treatment.

For other treatments for CHE, the efficacy of pegylated interferon (Peg-IFN) has been reported [84,85]. However, the number of cases investigated for the effect and safety has been small, and Peg-IFN may cause immunological adverse events. Although Peg-IFN can be used in liver transplant patients and non-transplant patients with CHE [84], Peg-IFN-alpha increases the risk of acute rejection in other transplant patients with CHE [86]. Regarding sofosbuvir (SOF), it was reported that SOF monotherapy had only modest antiviral efficacy in CHE patients and failed to achieve viral elimination [87]. Additional effects of SOF in combination with RBV have also not yet been established in vivo. However, a synergistic effect of 2′-methylcytidine or 2′-C-methylguanosine with RBV in inhibiting HEV replication in a reporter assay and cultured cells was recently reported [88]. Due to a lack of information demonstrating the efficacy and safety these medicines in clinical practice, large-scale clinical trials should be conducted in the future.

At present, RBV is the first-choice antiviral agent to treat CHE in both immunocompromised and immunocompetent cases. It is essential that the WHO continues to include RBV in the List of Essential Medicines, and the development of additional therapies is crucial [2].

As mentioned in Section 2, HEV infection can cause some disorders in other organ systems. Neurological disorders such as Guillain–Barré syndrome associated with HEV infection have been reported [28]. Antibodies that are produced against gangliosides through molecular mimicry after culprit infections have been shown to lead to Guillain–Barré syndrome. As a severe case of Guillain–Barré syndrome successfully treated with intravenous immunoglobulin has been reported [27], treatments by immunological approaches could be preferable to antiviral therapy in such patients. In addition, glomerulonephritis cases such as membranoproliferative glomerulonephritis, IgA glomerulonephritis, and membranous nephropathy caused by HEV infection have also occasionally been reported [29,30,31,89]. The patients with spontaneous HEV clearance subsequently show improved glomerulonephritis and renal function as well [30,31]. On the other hand, in patients who develop CHE, glomerulonephritis may also continue. Antiviral therapy with RBV is useful not only for the eradication of HEV, but also for the remission of glomerulonephritis and renal function in such patients [29,31,89]. Both neurological disorders and renal disorders may be attributed to the direct cytopathic effects of HEV and/or immune-mediated mechanisms. Although the question of whether antiviral therapy with RBV or immunological therapies such as corticosteroids or immunoglobulin should be preferentially given is still controversial, treatment should be selected considering the pathophysiological mechanisms, titer of HEV RNA, liver function, and the severity of neurological and/or renal disorder in each patient. Most acute pancreatitis cases associated with HEV have been reported from the Southern Asia region, suggesting that these were possibly caused by HEV genotype 1 infection [26]. It is possible that genotype 1 has high tropism for the pancreas, because no cases of acute pancreatitis have been reported in patients infected with other HEV genotypes [26]. Bazerbachi F et al. reported that of the 53 acute pancreatitis patients associated with HEV infection, 44 patients (83%) had mild and 9 (17%) had moderate–severe acute pancreatitis [26]. Although acute pancreatitis associated with HEV usually resolves with supportive treatments, early diagnosis may help in reducing morbidity and mortality [26]. Thus, clinicians should consider the complication of acute pancreatitis in patients with severe abdominal pain in the course of HEV infection. Although mild thrombocytopenia may coexist with HEV infection, it does not generally require any specific treatments [32].

### 3.5. Prognosis of CHE

The prognosis of CHE depends on the primary disease complicating immunosuppression, such as organ transplant, positive HIV, cancers (e.g., hematological malignancies), autoimmune diseases (e.g., rheumatoid arthritis, usually treated with immunosuppressants such as steroids, tacrolimus, etc.), and so on. Because the first-choice treatment for CHE in such situations should be the dose reduction of immunosuppressants, the control of the primary disease would be more important. Secondarily, the impact of drugs for CHE such as ribavirin would influence the prognosis of CHE. The effect of ribavirin on CHE has been reported to be adequate—around 80% of the rate of sustained viral response, as described in the treatment section [77,90] and a meta-analysis of liver transplant recipients [78]. CHE could progress to cirrhosis [5,6] if the diagnosis and treatment are delayed. It must be obvious that the early recognition and diagnosis of CHE should be mandatory to obtain a good prognosis.

## 4. Conclusions

HEV is usually spread by the fecal–oral route due to contaminated water or food. Genotypes 1 and 2 can infect humans, but genotypes 3, 4, and (rarely) 7 circulate in several animal species and are recognized as zoonotic infectants and casual agents of CHE. Hepatitis E is routinely diagnosed by IgA or IgM response to HEV, but it is sometimes misdiagnosed due to rare low responsiveness to these antibodies. Direct detection of the virus by PCR would be desirable and should be disseminated in the near future. CHE has recently been recognized as a type of chronic liver damage induced by HEV—usually in immunocompromised patients, such as those with organ transplants, HIV-infected patients, and those receiving chemotherapy for malignancies. CHE can be easily misdiagnosed by the usual diagnostic methods of antibody response, such as anti-HEV IgM or IgA evaluations, because of the low antibody response in immunocompromised patients. HEV RNA levels should be evaluated in these patients, and appropriate treatments—such as ribavirin—should be administered to prevent progression to liver cirrhosis and liver failure. In addition, rare cases of CHE in immunocompetent individuals have also been reported, so care should be taken to avoid missing a diagnosis. The diagnosis and treatment of CHE should be considered and performed appropriately to decrease instances of hepatitis-virus-related deaths around the world.

## Figures and Tables

**Figure 1 microorganisms-11-01303-f001:**
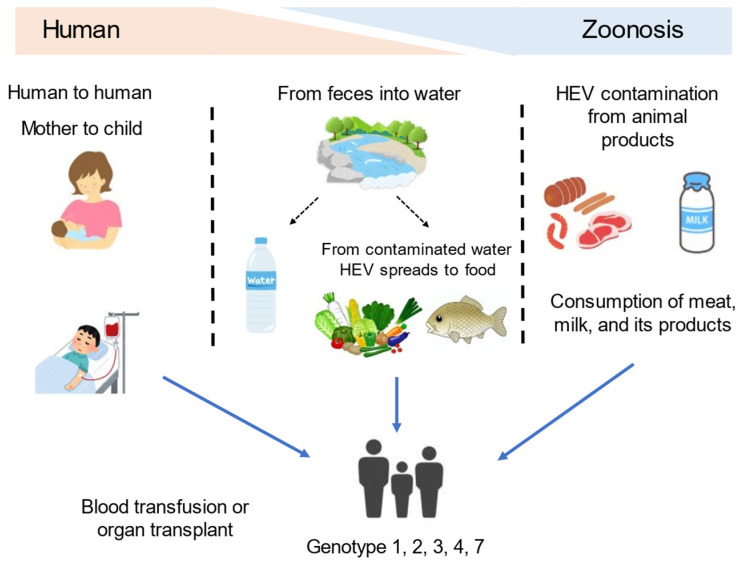
Transmission routes of the hepatitis E virus to humans.

**Figure 2 microorganisms-11-01303-f002:**
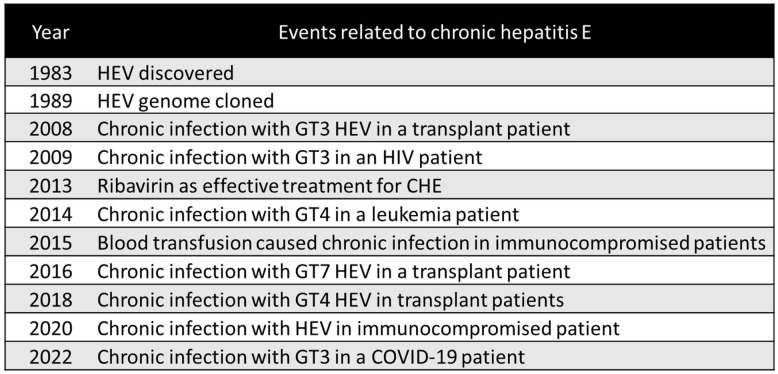
Historical events concerning chronic hepatitis E. HEV, hepatitis E virus; GT, genotype; HIV, human immunodeficiency virus.

**Figure 3 microorganisms-11-01303-f003:**
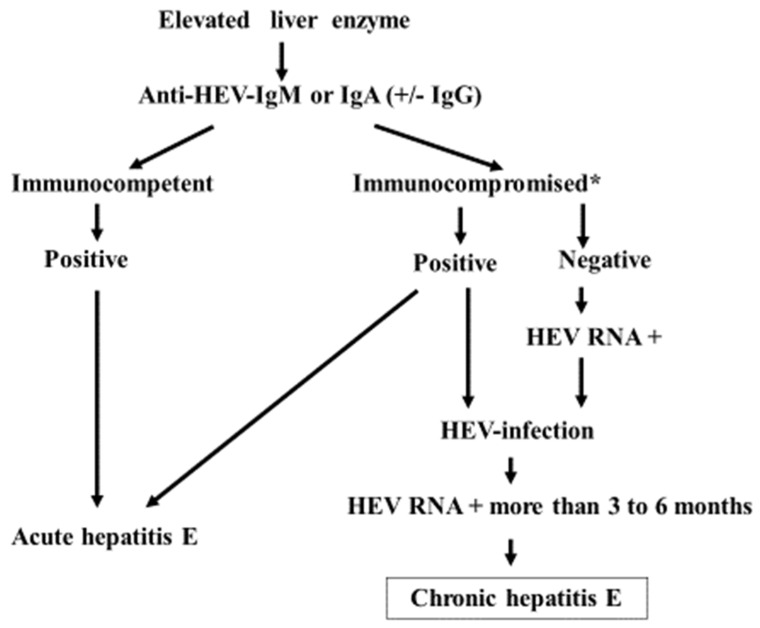
Diagnostic procedure of hepatitis E: The initial approach for diagnosing hepatitis E usually involves evaluating anti-HEV antibody, IgM, or IgA values. In patients with organ transplants, chemotherapy for hematologic malignancy, HIV, etc. (*), HEV RNA should be evaluated to confirm the true diagnosis, as false negatives can occur due to the poor antibody response.

**Table 1 microorganisms-11-01303-t001:** Early anti-HEV responses in chronic hepatitis E in Japan.

Case	Age (Years)	Sex	Underlying Disease	Immunosuppressant	HEV Sub-Genotype	Anti-HEV IgG	Anti-HEV IgM	Anti-HEV IgA	Reference
1	30	M	DCM, after heart transplantation	PSL, CyA, MMF	3b	−	−	+	[61]
2	33	M	CRF, after kidney transplantation	PSL, CyA, MMF	3a	+	−	+	
3	25	M	RN, after kidney transplantation	PSL, CyA, MMF	3a	+	+	+	
4	41	M	HCC, LC-NASH, after liver transplantation	PSL, tacrolimus, MMF	3b	+	+	−	[62]
5	37	F	Complete remission of Burkitt lymphoma	Rit, CPA, VCR, DXR, Ara-C, DEX	3	N.D.	N.D.	−	[63]
6	60	F	PBC, after liver transplantation	tacrolimus, MMF	3	+	N.D.	N.D.	[64]
7	76	F	Hypertension, dyslipidemia	PSL, AZP	3a	+	+	−	[54]

Ara-C, cytarabine; AZP, azathioprine; CPA, cyclophosphamide; CRF, chronic renal failure; CyA, cyclosporine A; DCM, dilated cardiomyopathy; DEX, dexamethasone; DXR, doxorubicin; HCC, hepatocellular carcinoma; HEV, hepatitis E virus; IgA, immunoglobulin A; IgG; immunoglobulin G; IgM, immunoglobulin M; LC, liver cirrhosis; MMF, mycophenolate mofetil; NASH, non-alcoholic steatohepatitis; N.D., not described; PBC, primary biliary cirrhosis; PSL, prednisolone; RN, reflux nephropathy; Rit, rituximab; VCR, vincristine.

**Table 2 microorganisms-11-01303-t002:** Prevention and treatment of HEV.

Patients/Recipients		Recommended Treatment
Patients on the waiting list for organ transplantation Pregnant women living in endemic regions		Vaccination
Immunocompromised CHE patients		Reduction in immunosuppressive therapy RBV
Immunocompetent CHE patients		RBV
Extrahepatic manifestations	Neurological disorders	Intravenous immunoglobulin, corticosteroids, RBV
	Glomerulonephritis associated with CHE	RBV, corticosteroids
	Acute pancreatitis	Supportive treatments
	Thrombocytopenia	No any specific treatments

CHE, chronic hepatitis E; HEV, hepatitis E virus; RBV, ribavirin.

## Data Availability

All data generated or analyzed during this study are included in this published article.

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
