# Peer review of "The Diagnosis, Pathophysiology, and Treatment of Chronic Hepatitis E Virus Infection—A Condition Affecting Immunocompromised Patients"

_microorganisms, 2023, doi:10.3390/microorganisms11051303_

Round 1

Reviewer 1 Report

This review summarizes the current hepatitis E virus infection including the susceptible population, the current diagnostic methods and treatment methods. To a certain extent, the review has some enlightenment for future liver disease research.

However, some major concerns are needed to be clarified. As follows:

1. About HEV infection diagnosis: When is the specific time for HEV antibody IgM, IgG IgA detection? the time of HEV RNA detection?

2. HEV infection is a self-limited course. When is the detection time and the positive rate of various antibodies high?

3. And what is the specific basis of diagnosis not clearly stated?

4. What is the specific dose and choice of antiviral drugs for normal and immunocompromised patients?

5. In the conclusion part, there is no summary of the infection process and current situation of HEV infection, no specific diagnosis methods at present, and no summary of the current situation and future research direction of hepatitis HEV.

Author Response

May 3, 2023

Re: Manuscript No. microorganisms-2305292

Dear Ms. Essie Liang,

Editorial office of Microorganisms.

Thank you very much for your letter of April 24, 2022 and for the reviewer’s constructive and insightful comments on our manuscript # microorganisms-2305292. We have made the changes to accommodate the reviewer’s comments. We followed all your insightful suggestions. The revised manuscript improved following reviewer’s comments.

Reviewer #1:

This review summarizes the current hepatitis E virus infection including the susceptible population, the current diagnostic methods and treatment methods. To a certain extent, the review has some enlightenment for future liver disease research. However, some major concerns are needed to be clarified. As follows:

  1. About HEV infection diagnosis: When is the specific time for HEV antibody IgM, IgG, IgA detection? the time of HEV RNA detection?

Answer; Thank you for suggestion. We added the description about HEV antibody IgM, IgG, IgA and HEV RNA for diagnosis and specific times for HEV infection. Line 92 to line 109.

  1. HEV infection is a self-limited course. When is the detection time and the positive rate of various antibodies high?

Answer; Thank you for suggestion. We added the description about detection time of HEV antibody and HEV RNA for diagnosis of HEV infection. We also mentioned the differences in specificity between the IgM class HEV antibody measurement system and those of IgA measurement system. Line 92 to line 109.

  1. And what is the specific basis of diagnosis not clearly stated?

Answer; Thank you for suggestion. We added the description about HEV antibody and HEV RNA for diagnosis and detection times for HEV infection. Line 92 to line 109.

  1. What is the specific dose and choice of antiviral drugs for normal and immunocompromised patients?

Answer; Thank you for your comment. The previous studies showed the excellent outcome of RBV at the median dose of 600 mg/day and other treatment selection have not been established in both immunocompromised and immunocompetent patients. We added the description for that in the paragraph regarding treatments. Line 358 to line 361

  1. In the conclusion part, there is no summary of the infection process and current situation of HEV infection, no specific diagnosis methods at present, and no summary of the current situation and future research direction of hepatitis HEV.

Answer; Thanks for your suggestion. I added the infection process, specific diagnostic methods, current situation and research direction of HEV in the conclusion. Line 442 to line 447.

We believe that because of the reviewers’ insightful comments, we were able to significantly improve and clarify our manuscript, and trust that the revised manuscript will meet your approval.

Sincerely yours,

Satoru Kakizaki, M.D., Ph.D., AGAF.

Department of Clinical Research, National Hospital Organization Takasaki General Medical Center, 36 Takamatsu-cho, Takasaki, Gunma 370-0829, Japan.

Tel: +81-27-322-5108, Fax: +81-27-322-6111

Reviewer 2 Report

Interesting paper, focusing a neglected topic.

I have only one suggetion:

Please, add at the end of the title the sentence " ...., a condition seldom affecting immunocompromised patients." . Otherwise the title may result misleading.

Author Response

May 3, 2023

Re: Manuscript No. microorganisms-2305292

Dear Ms. Essie Liang,

Editorial office of Microorganisms.

Thank you very much for your letter of April 24, 2022 and for the reviewer’s constructive and insightful comments on our manuscript # microorganisms-2305292. We have made the changes to accommodate the reviewer’s comments. We followed all your insightful suggestions. The revised manuscript improved following reviewer’s comments.

Reviewer #2:

Interesting paper, focusing a neglected topic. I have only one suggestion: Please, add at the end of the title the sentence " ...., a condition seldom affecting immunocompromised patients.". Otherwise, the title may result misleading.

Answer; Thank you for suggestion. We added the sentence following your suggestion. Line 2 to line 4.

We believe that because of the reviewers’ insightful comments, we were able to significantly improve and clarify our manuscript, and trust that the revised manuscript will meet your approval.

Sincerely yours,

Satoru Kakizaki, M.D., Ph.D., AGAF.

Department of Clinical Research, National Hospital Organization Takasaki General Medical Center, 36 Takamatsu-cho, Takasaki, Gunma 370-0829, Japan.

Tel: +81-27-322-5108, Fax: +81-27-322-6111

Reviewer 3 Report

Comments to Authors:

The review by Satoshi Takakusagi et al. is a narrative review regarding diagnosing and treating the hepatitis E virus (HEV). 

This review is well done and only minor changes are necessary to be accepted into Microoganisms

Minor comments:

1.     Title. Authors could change “chronic hepatitis E” to “hepatitis E virus”

2.     Abstract. Line 17. Please include information about the type of hepatitis “…been reported as CHE”

3.     Introduction. Line 38. Elevation of liver enzymes is not a symptom

4.     Lines 40-43 are similar to lines 86-88. Please change the sentence to not repeat information

5.     Hepatitis E virus section. Genotypes 1-2 and 3-4 are different illnesses (only genotypes 3 and 4 are zoonosis) Please explain in different paragraphs. Please, include information regarding genotype 7. Please include the type of “genotypes other than 1 and 2. 

6.     Treatment for CHE. Authors could change the title to “Prevention and treatment of HEV” and include in this section information about the vaccine. Please include the information about extrahepatic manifestations treatment in this section and include more information. Finally, treatments could be summarized as a new Table

7.     Please include a new paragraph regarding the prognosis of CHE and prognostic factors. 

8.     Please include important references such as Journal of Hepatology 2019 vol. 71:465–472; Journal of Hepatology 2020 vol. 72:1105–1111; Viral Hepat. 2022 vol. 29:698–718; and J Viral Hepat. 2023; vol. 30:101–107.

Author Response

May 3, 2023

Re: Manuscript No. microorganisms-2305292

Dear Ms. Essie Liang,

Editorial office of Microorganisms.

Thank you very much for your letter of April 24, 2022 and for the reviewer’s constructive and insightful comments on our manuscript # microorganisms-2305292. We have made the changes to accommodate the reviewer’s comments. We followed all your insightful suggestions. The revised manuscript improved following reviewer’s comments.

Reviewer #3:

Comments to Authors:

The review by Satoshi Takakusagi et al. is a narrative review regarding diagnosing and treating the hepatitis E virus (HEV). This review is well done and only minor changes are necessary to be accepted into Microoganisms.

Minor comments:

  1. Title. Authors could change “chronic hepatitis E” to “hepatitis E virus”

Answer; Thank you for suggestion. We added the word “virus” following your suggestion. Because we want to use “chronic”, we added the word “infection” as “chronic hepatitis E virus infection”. Line 2 to line 4.

  1. Abstract. Line 17. Please include information about the type of hepatitis “…been reported as CHE”

Answer; Thank you for suggestion. We corrected it following your suggestion.

  1. Introduction. Line 38. Elevation of liver enzymes is not a symptom.

Answer; Thank you for suggestion. We corrected it.

  1. Lines 40-43 are similar to lines 86-88. Please change the sentence to not repeat information.

Answer; Thank you for suggestion. We corrected it.

  1. Hepatitis E virus section. Genotypes 1-2 and 3-4 are different illnesses (only genotypes 3 and 4 are zoonosis). Please explain in different paragraphs. Please, include information regarding genotype 7. Please include the type of “genotypes other than 1 and 2.

Answer; Thank you for your comment. We summarized the genotypes and modified the Figure 1. We also included the information regarding genotype 7. Line 59 to line 73.

  1. Treatment for CHE. Authors could change the title to “Prevention and treatment of HEV” and include in this section information about the vaccine. Please include the information about extrahepatic manifestations treatment in this section and include more information. Finally, treatments could be summarized as a new Table.

Answer; Thank you for your comment. We changed the title from “Treatment for CHE” to “Prevention and treatment of HEV” according to your advice. Further, the information about vaccine and extrahepatic manifestations treatment were added and we made new Table for Treatment for CHE. Line 324 to line 338. Line 390 to line 420. New table 2.

  1. Please include a new paragraph regarding the prognosis of CHE and prognostic factors.

Answer; We added the prognostic section as 3.5. Line 427 to line 438.

  1. Please include important references such as Journal of Hepatology 2019 vol. 71:465-472; Journal of Hepatology 2020 vol. 72:1105-1111; J Viral Hepat. 2022 vol. 29:698-718; and J Viral Hepat. 2023; vol. 30:101-107.

Answer; Thank you for suggestion. We included these references.

We believe that because of the reviewers’ insightful comments, we were able to significantly improve and clarify our manuscript, and trust that the revised manuscript will meet your approval.

Sincerely yours,

Satoru Kakizaki, M.D., Ph.D., AGAF.

Department of Clinical Research, National Hospital Organization Takasaki General Medical Center, 36 Takamatsu-cho, Takasaki, Gunma 370-0829, Japan.

Tel: +81-27-322-5108, Fax: +81-27-322-6111
